# Assessing Extensive Semi-Arid Rangeland Beef Cow-Calf Welfare in Namibia. Part 2: Categorisation and Scoring of Welfare Assessment Measures

**DOI:** 10.3390/ani11020250

**Published:** 2021-01-20

**Authors:** Yolande Baby Kaurivi, Richard Laven, Rebecca Hickson, Tim Parkinson, Kevin Stafford

**Affiliations:** 1School of Veterinary Medicine, Massey University, Private Bag 11 222, Palmerston North 4442, New Zealand; r.laven@massey.ac.nz (R.L.); t.j.parkinson@massey.ac.nz (T.P.); 2School of Agriculture and Environmental Management, Massey University, Private Bag 11 222, Palmerston North 4442, New Zealand; R.Hickson@massey.ac.nz (R.H.); k.j.stafford@massey.ac.nz (K.S.)

**Keywords:** animal welfare assessment, categorisation, beef cow systems, semi-arid rangelands, Namibia

## Abstract

**Simple Summary:**

Basing welfare assessment standards on protocols developed for industrialised beef exporting countries could enhance the beef production and export standards in Namibia. This paper compares imposed thresholds of welfare measures of categorisation and derived thresholds to see—which was the most appropriate to the range of observations and welfare implication in three cow-calf production systems in Namibia. Using the same thresholds as the New Zealand protocol, regardless of the farm system, commercial herds achieved most welfare measures thresholds, but semi-commercial and communal herds attained acceptable welfare thresholds only in a few measures. For measures with significant welfare implications, the stricter threshold was retained, while derived thresholds appeared more appropriate for commonly occurring traits (but of less welfare importance), and some measures threshold were temporarily adjusted to reflect drought conditions. The welfare assessment identified the strengths and weaknesses in thresholds in measures across the farm types, which is envisioned to draw attention for remedial intervention to improve welfare standards of the beef industry.

**Abstract:**

This paper aims to develop standards for a welfare assessment protocol by validating potential categorisation thresholds for assessing beef farms in various beef cow-calf production systems in Namibia. Forty measures, combined from a New Zealand-based protocol plus Namibia-specific measures, are applied to 55 beef farms (17 commercial farms, 20 semi-commercial and 18 communal village farms) during pregnancy testing, and a questionnaire guided interview. The categorised measures on a 3-point welfare score (0: good, 1: marginal, and 2: poor/unacceptable welfare) are subsequently compared with the derivation of thresholds based upon the poorest 15% and best 50% of herds for each measure. The overall combined thresholds of continuous measures across the three farm types show 10/22 measures that posed welfare compromise across Namibia, whereas commercial farms have 4/22 measures, and semi-commercial and communal village farms have 12/22 and 11/22, respectively, with high thresholds. Most measures-imposed thresholds are retained because of significant importance to the welfare of animals and preventiveness of the traits, while leniency was given to adjust good feeding and mortality measures to signify periods of drought. Handling measures (fearful, falling/lying) and abrasions thresholds are adjusted to reflect the temporary stress caused by infrequent cattle handling, and faulty yard designs/design and possible cattle breed influence on handling. Hence, Namibia needs prioritised investigation of underlying contributing factors and remediation to reduce the high thresholds.

## 1. Introduction

Animal welfare in Namibia has been increasingly acknowledged as important in the production and trade of farm animals and their products. International trading partners for Namibia’s beef are seeking assurances that acceptable animal welfare standards are maintained. This requires concrete animal welfare legislation, including a standardised, validated protocol for assessing beef cattle welfare. Currently, the Farm Assured Namibian Meat (FAN Meat) scheme provides assurance that Namibian beef is produced under a natural, safe environment and is traceable and of good quality, but the scheme is not comprehensive as a welfare assessment protocol [1]. For example, key aspects of the FAN Meat animal welfare standards pertain to livestock keeping and transportation, and state that animal handling facilities must be adequate to ensure ease of handling and to prevent injuries to animals but does not provide specific recommendations or determine thresholds. The significant thresholds relevant to animal welfare covered under the FAN Meat scheme are those related to dehorning and castration, which require that these procedures are undertaken in animals younger than two months of age.

Currently, welfare standards in Namibian cattle are only routinely assessed at export abattoirs [2] with no standardised on-farm assessment. A validated, comprehensive, farm-based animal welfare assessment protocol with thresholds comparable to those used in other beef exporting countries would, thus, be useful for Namibia.

On-farm welfare data from both categorical measures (i.e., recorded at the individual farm level as good, marginal and poor welfare) and continuous measures (recorded as the percentage of cattle affected on each farm) need meaningful interpretation to reflect the welfare status of farms [3]. The results of categorical measures provide clear indications for farmers and their advisors regarding future actions: Good welfare—continue current practices; poor welfare—remedial action is needed; marginal welfare—assess current practice and aim for improvement [4]. The raw results for continuous measures do not provide such clear indications; they may be useful for benchmarking, but unless thresholds are identified at which action should be taken [5], farmers and their advisors may simply focus on prevalence as a means for determining actions rather than whether the prevalence is acceptable or not. Putting achievable thresholds across the prevailing beef production systems is critical for the country to benchmark and apply adequate remedial strategies for identified welfare compromise [6,7].

Using Namibian data we collected in Part 1 of the study [8], the present paper aimed to replicate the process used by us as part of the development of an animal welfare assessment protocol for extensive beef systems in New Zealand [4]. This is done to develop thresholds which could be used to categorise continuous animal welfare measures into acceptable, marginal and non-acceptable welfare. This aim was achieved through the setting and comparing of imposed categorisation thresholds with derived thresholds to see, across the farm systems in Namibia, which were the most appropriate thresholds to use to identify poor welfare and to stimulate improvement.

## 2. Materials and Methods

### 2.1. Description of the Study Areas

The study areas represent their farm types within Namibia, as described by Kaurivi [8]. In summary, commercial farms (*n* = 17 farms/herds) (Gobabis area, Omaheke region, eastern Namibia) were large privately-owned farms (range 3000–10,000 ha) that were individually fenced off in several grazing paddocks. Semi-commercial village farms (*n* = 20 herds) in Okakarara area (Otjozondjupa region) were government-owned settlement villages predominantly inhabited by farmers who were dependent on livestock production for their livelihood. Multiple families’ (~10–30 families) cattle grazed and shared water on the same permanent communal land with limited or no internal and external border fencing. These farms were eligible to produce beef for export markets. Communal village farms (*n* = 18 herds) in Opuwo/Kaokoland area, (Kunene region, north-western Namibia) were exclusive government-owned communal settlements with inhabitants farming with small multipurpose (i.e., milk, meat, dung for building) cattle holdings mainly for subsistence. As on semi-commercial village farms, multiple families’ (~10–30 families) cattle grazed and shared water on the same permanent communal land with limited or no internal and external border fencing. Communal village farms were in an area restricted for animal disease control to produce beef for overseas export markets. 

### 2.2. Welfare Assessments

The protocol used is described in full in Kaurivi [9]. The welfare assessment took place in March/April 2019 (autumn) on 55 farms during yarding of 2529 cows for pregnancy testing. In the yards, observations were made in the race (or pens) of body condition, rumen-fill, physical health, presence of long/sharp horns and behaviour. Stockpersonship was evaluated as cows entered, were handled and exited the race (or, if no race, when cows were restrained for pregnancy testing in pens). The yard design and handling facilities were also evaluated for ease of handling of cows. A farm resource evaluation and a questionnaire guided assessment of health and management of the herd in the preceding 12 months was undertaken, i.e., castration, dehorning, cattle identification practices, vaccinations, cattle deaths and disease incidences.

### 2.3. Data Analysis

All data were analysed using IBM SPSS Statistics for Windows Version 24 (IBM Corp. Released 2016. Armonk, New York, USA). Descriptive statistics for continuous measures were used to capture central tendency (median), and range (minimum and maximum). Qualitative methods were used to analyse the frequency of ordinal measures. The Shapiro-Wilk test was used to test for normality (significance level *p* ≤ 0.05), and log10^(n+1)^ was used to transform those variables that were not normally distributed.

### 2.4. Categorisation and Refinement of Measures

The categorisation of measures was principally based on the New Zealand protocol on a 3-point welfare score of 0: Good welfare (no intervention necessary, but monitoring); 1: Marginal welfare (plan for intervention, increase monitoring); and 2: Poor/unacceptable welfare (immediate intervention required) [4]. The initial categorisation of welfare thresholds in the New Zealand protocol was based upon the authors’ perception of acceptable welfare standards and the consensus of the literature. Subsequently, an alternative approach to applying pre-determined value judgements was to determine the threshold from the data, so that an arbitrary 15% of herds were considered poor and 50% good. Derived thresholds were determined based on z scores to result in approximately 50% of herds falling into an acceptable welfare band (‘green’) and 15% of herds into a poor welfare band (‘red’). Herds not in the green or red band were classified as orange. The 15% was chosen to fit with the ‘15% rule’ where animals (in this case, herds) below this point are considered as worse-off in terms of animal welfare compromise [10,11]. For each non-categorical measure, the derived red threshold and the imposed score 2 threshold were then compared by dividing the derived threshold by the imposed threshold.

The New Zealand thresholds were compared with the pre-determined thresholds of each continuous measure in the current data from the three farming systems in Namibia. For example, for good health measures (i.e., blindness, mortality), painful conditions (e.g., dystocia) and absence of injuries or physical impairment the welfare thresholds were kept the same at 2%. For the additional Namibia-specific measures, the thresholds for extraneous (multiple) brands/cuts (5%), long/sharp horns (10%) and external parasites (10%) were based on authors’ opinion of acceptable welfare standards (no concrete thresholds could be found in literature). The threshold (10%) for tail twisting was aligned with that of hitting of cows. The distance to water and grazing categorisation was based on Holechek [12] and the use of electrical prodders (<1%) on Grandin [13]. The imposed thresholds were applicable for all farm types; there was no division between village farms or from commercial farms. Farms/herds were not given an ‘overall’ welfare score [14] because all of the measures were considered important, and an overall acceptable score may mask unacceptable welfare in some measures [5]. Categorisation thresholds and details of how each measure in the Namibia protocol was assessed are presented in Table 1, Table 2 and Table 3.

## 3. Results

### 3.1. Categorisation Results

Categorised observational data are illustrated in Figure 1 (measures of feeding, and environmental factors), Figure 2 (health indicators), Figure 3 (frequencies of painful procedures), Figure 4 (animal-based stockpersonship scores) and Figure 5 (resource and management-based stockpersonship scores).

Feeding measures: Cattle on commercial farms did not walk long distances to grazing and water (<1.6 km–3.2 km), and there were no emaciated cows observed, and 76% herds were in good body condition, and 88% of cows had good rumen fill. Semi-commercial and communal herds obtained a good welfare score for BCS only in 40% and 17% of herds, respectively, of which most cattle herds in communal villages were classified as more emaciated (<2/5 scale) than just thin. Cattle walked long distances to water and grazing (>3.2 km), and all village farm herds scored poorly in this. Hazards were mostly marginal across the farm types, with semi-commercial and communal showing 35% and 33% of herds with poor welfare score.

Health observations: The worst scores were observed for abrasions and mortality at all farm types. Only 35% of commercial and 25% of semi-commercial herds had a good score for abrasions, and none of the communal herds had a good score for abrasions. The mortality rate was categorically poor in 65%, 95% and 100% of commercial, semi-commercial and communal herds, respectively. Fifty percent of semi-commercial herds had a good score for dystocia, while only 24% and 33% of commercial and communal herds did, respectively. Lameness and extraneous brands/cuts were worst in communal followed by semi-commercial herds. The tick and fly burden were worse in semi-commercial with 20% and 40% herds with a poor score, and 17% and 11% of communal herds had poor welfare for ticks and flies, respectively, while the majority of commercial herds had a good score for both ectoparasites types.

Painful management measures: Procedures were performed without the use of anaesthesia or analgesics. All farm types identified cattle with ear tags (marginal welfare), while most herds at the village herds had poor scores for ear notching (with knives) and extraneous branding/cuttings. In relation to disbudding, those commercial herds who disbud, the mode for age was 2 months, while it was 6 months at the village herds. Only 29% of herds had cows with long/sharp horns in commercial compared to 95% and 100% for semi-commercial and communal herds, respectively. Communal village farms had the most herds with poor scores for dehorning, castration, and ear tagging/notching.

Animal-based stockpersonship measures: Scores were similar across the farm types, except for electrical prodders, which were only used at 5/17 commercial herds, producing a poor score at those herds (>1% cows prodded). Commercial farms had the least herds with poor welfare for fearful/agitated, hitting and tail twisting. All the farm types scored poorly (>2% threshold) for falling/lying.

Resource-and management-based stockpersonship measures: Good welfare score was showed for equipment noise and dog noise and health checks, at most herds across the farm types. Commercial farms had less noisy handlers and more herds with >4 times yarding frequency per year (good welfare), and effective yard handling/flow. The communal (95% of herds) cattle yarding frequency/year (2 or fewer times) was the poorest than the other farm types.

### 3.2. Refined Thresholds

Derived threshold values for the three farm types are shown in Table 4 For commercial, of the 22 measures, only falling/lying was normally distributed. The measures that were normally distributed for semi-commercial herds were thin and poor rumen fill cows, long/sharp horns, and mortality rate. In addition to these latter measures, abrasions, falling/lying and fearful/ agitated cows’ measures were also normally distributed for communal herds. Commercial herds had 4/22 measures (abrasion, mortality, fearful and falling/lying) with a derived red threshold that were >2 times the imposed threshold, with ratios ranging from 2.5 (fearful/agitated) to 6.9 (falling/lying). Including these measures, both semi-commercial and communal herds had a similar total of 12/22 and 11/22 measures, respectively, with a derived red threshold that was >2 times the threshold imposed by categorisation. The measures were related to good feeding (thin, emaciated and poor rumen fill cows), extraneous brands/cuts, long/sharp horns, swelling and hair loss, and as well as dystocia at semi-commercial herds and lameness at communal herds. All three farm types had similar red thresholds (~12%) for cows falling/lying during handling. The highest red threshold was thin cows for communal (97.7%), whilst the proportion of emaciated cows was 82.9% for communal herds.

All combined derived thresholds for the 55 beef cow herds are shown in Table 5. In this regard, 10/22 measures had derived red threshold that were >2 times the imposed threshold with ratios ranging from 2.4–16.6%. These included all the four measures where commercial herds failed (abrasion, mortality, fearful and falling/lying) and in addition, three measures related to good feeding (thin, emaciated, poor rumen fill) and swelling, hair loss and long/sharp horns.

## 4. Discussion

Kaurivi [8] compared the effects of farm production systems on the welfare of beef cows with results showing better welfare on commercial, followed by semi-commercial, and the communal herds in the least. The current paper used that data to develop standards for a welfare assessment protocol, by validating potential thresholds which could be used to categorise continuous animal welfare measures into acceptable, marginal and unacceptable welfare for assessing beef farms in various cow-calf production systems in Namibia. The categorised measures scores showed a marked separation of commercial farms from semi-commercial and communal village farms, but no separation between the village farm types. The same thresholds were used for all farm systems, regardless of the village herds not achieving imposed thresholds that are achievable at commercial herds, because the impact of a situation or condition on the welfare of the cow is not mediated by the type of farm she lives on. The setting of thresholds should reflect standards of “what should be and not what is”, and common findings that are compromising the welfare of animals should not be accepted as acceptable standards [17]. Critically, multiple observers would ensure better reliability of scoring of the measures, and hence, confidence in the validity of the results, than the imminent subjectivity that may be caused by one person.

The findings indicate that to improve most of the compromised traits, development of interventions should be geared towards what works in village farm herds. Whereas for some other traits, there are herds exceeding the threshold in all herd types, and solutions must be developed for all herd type. Although the long term target is to have improved derived threshold in all measures at all the farms (a target of at least 50% of the herds in good welfare), an initial target to improve the worst 15% of herds might be the logical first step, and then revise the red thresholds, to gradually to pull the recovering herds to better. It is risky to set a target that 90% of the herds currently fail; this may deter farmers from making improvements and instead dismiss the target as unrealistic. See Appendix A (Table A1) for the summary of continuous welfare measures scoring threshold for Namibia beef cattle systems compared to New Zealand thresholds.

With regards to *good feeding*, with below-average rainfall (since 2013) across Namibia, the 2019 drought (the driest year in 90 years) was treated as a state of emergency by the government [18]. The impact of drought had more dire consequences on the village farm types (i.e., high thin and poor rumen thresholds and long-distance walking to grazing) than for commercial farms, because of the challenges of communal grazing land without proper demarcation that does not allow sustainable rangeland management [19,20]. In terms of identifying the need for nutritional intervention, stringent thresholds for low BCS (between 5–15%) was supported [21,22]. This should even be truer for emaciated cows (BCS < 2/5 scale) that indicate a greater unacceptable compromise of cattle welfare [15]. In New Zealand, recommendations are that cows’ body condition not to be below 4/10 scale [23] and under Australian legislation, it is not acceptable for producers to allow animals to die because of starvation and thirst [24]. For Namibia, an adjustable lenient threshold setting could be climate-responsive, i.e., in a drought, cows are going to get thinner than a normal year, and a farmer who has thin cows in a good year is a worse farmer than a farmer who has the same prevalence of thin cows in a drought, but the level of emaciation in cattle is unacceptable regardless. Taken together, an adjusted threshold to reflect drought conditions could be 30% (orange thresholds 29.4%), while retaining the imposed threshold of 10% during sufficient rainfall periods. Nonetheless, drought in Namibia is predicted to be more frequent than before, due to the impact of climate change [20]. Hence, there is a need to put mitigating strategies in place (i.e., supplementary feeding, targeted livestock grazing, and timely adjustments to animal numbers; [25,26], especially in the village farms. Similar adjustments could also apply to poor rumen fill (RFS). However, although RFS is a good indicator of nutritional status of cows, it reflects animals recent feed intake, thus, the low imposed threshold of 50% is appropriate for the detection of poor nutrition of extensively reared cows that are likely to be drafted a day before an assessment. This short-term deprivation of feed might be true during normal rainy seasons across Namibia beef farms, but it reflected a long-term underfeeding, due to the forage shortage at the village farms. Distinguishing between short and long-term poor RFS was impractical, hence, the imposed threshold was retained. Over a long period of time, poor RFS could bring awareness and correction of compromising feed deprivation or other underlying problems.

Regarding *appropriate environmental issues*, dirtiness and diarrhoea were the only measures where the thresholds (both thresholds were 20%) fit for Namibia were reduced. In New Zealand, diarrhoea was faecal soiling, due to the high-water content in the pasture. Thus, a 5% threshold (between 3% and 6% orange and red threshold, respectively) fit for Namibia where grass grows seasonally and between years, and where diarrhoea cases may also be of pathological nature (i.e., infectious causes, internal parasites, and plant toxicity) was essential. Similarly, a 5% threshold was more fitting for dirtiness, where cattle on drier pasture are less dirty; red thresholds were only up to 3.6% in this study (Table 4). Moreover, stringent thresholds might also not be relevant for dirtiness, a welfare measure that is questionable for assessing cattle in an extensive system [4,27], unless cattle were destined for slaughter [13].

For most *health-related measures*, the imposed thresholds were retained despite the high occurrence of traits, considering the welfare implications to animals, possible preventative and mitigating strategies available. For painful conditions like lameness and dystocia, a 2% threshold was retained and is attainable on Namibia beef herds (mean 1.9% and 1.7%, respectively). For measures related to skin alterations (swelling, hair loss and abrasion) the derived thresholds were more than twice the imposed threshold for the combined herds, indicating high prevalence. The prevalence of swelling was attributed to faulty injections; thus, there is room for improving the skills of injections across all farm types, through awareness and training. Similarly, the high derived thresholds of hair loss at the village herds could have been prevented with preventative vaccination of the endemic lumpy skin disease in the country. Hence, an imposed threshold of 2% was also retained as fitting for these conditions. On the other hand, the much higher prevalence of abrasions on cattle across all the beef farm types is indicative of a welfare concern in the country, so an initial target of 2% would likely be unrealistic. However, abrasions are painful and cause suffering for the animal. Efforts should be directed toward identifying the causes of the abrasions and attempting to remove relevant hazards from the environment, such as faulty yard designs and yard structural materials [27]. Given 59–96% of herds in the different herd types exceed the imposed threshold, a higher threshold (e.g., 7%; orange threshold) could be applied initially to allow improvements to be made in the worst affected herds first.

Setting welfare thresholds for any measures should reflect the significance of that trait to the welfare of animals and not necessarily according to the status quo. A good example was the mortality rate that varies in extensive beef systems with the type of production system, the location, and herd management [28,29]. In this study, cattle losses indicated a worrisome occurrence in all farm types (derived thresholds >9 times the imposed thresholds). The mortality rate on commercial herds at an average of 3.4% (range 0–11.5%) was consistent with an average of 3.5% cattle losses in another commercial herd in Namibia over a 13 year period [29]. These figures are similar to cattle losses (average 3.9%) at 25 beef farms in New Zealand, where the feasibility of categorisation thresholds used in this study was verified [30]. However, an imposed threshold of 2% was retained in that protocol, given the significance of the measure to the welfare of cattle [4], and the same threshold was, thus, taken for Namibia. Judging from the mortality rate median of 2.4% at commercial herds, there is potential for attaining the imposed thresholds. However, like for feeding measures, an adjusted lenient threshold to reflect drought condition (i.e., increased predation, poor nutrition effects), that contributed to cattle losses could be 7% (orange thresholds of combined herds). It will be interesting and useful to validate the mortality threshold at farms during years of average rainfall. Setting mortality threshold for Namibia beef production systems is an important step towards creating awareness of the risks of cattle losses on the farm towards, strategising remedial actions, and thus, a stringent threshold is worth keeping.

Ectoparasites have a significant impact on the welfare of animals (i.e., transmit diseases, biting stress, skin injuries, and irritation) and a high incidence can markedly impair farm productivity [31,32]. However, the impact of different parasites on animals is different, i.e., cattle with high-level tick burden are more likely to be welfare compromised than those with fly burden. For example, ticks in Namibia transmit significant cattle disease, e.g., babesiosis, anaplasmosis, sweating sickness, and including lumpy skin disease, while flies are of a lesser impact [32,33]. Thus, the category imposed as an acceptable or nonacceptable compromise might have to vary depending on the welfare impact of each parasite type. In the present study, the number of ticks or flies on any animal to be regarded as a burden was the same at 20, and the threshold for unacceptable welfare was the same at 10%. The presence and thresholds setting of ectoparasites on extensive cattle has not been validated in animal welfare assessment protocols, and it is worthwhile validating our findings elsewhere in similar production systems to determine acceptable thresholds. For example, in northern Australia, treatment is only recommended for individual beef cattle infested with more than 200 buffalo flies [34]. Most of the predominant cattle breeds in Namibia are also tick tolerant [35], and ectoparasites burden is significantly influenced by seasonality and environmental factors [36]. Thus, thresholds might differ in countries, production systems and seasons. Nonetheless, a threshold of 10% (for both flies and ticks) is attainable across beef production systems with preventative herd health management. This was evident in the low mean incidence of both ticks and flies at commercial farms where ectoparasites treatment is commonly done.

With regards to *painful management procedures*, unlike castration and disbudding that can be performed at an early age (<2 months) and with anaesthesia to mitigate pain [37], the downside of branding early is that the brand becomes unreadable with time in adult cattle. Rebranding is a permissible requirement for cattle trading in Namibia [38], but the practice of multiple brands requires longer restraining of animals and causes more wounds on any part of the animals’ body [39]. Farms with only the initial brand were given a marginal score of welfare, while the extraneous brands/cuts (to mitigate the risk of stock theft) were regarded more extreme and indicated a poor welfare score. The village farms had derived red thresholds that were ≥3 times higher than their imposed thresholds for extraneous brands/cuts, indicating a trait with a high occurrence at these farm types. However, the imposed threshold of >5% of cattle in a herd with this trait (also close to the orange threshold of 3% and 4% at the villages) was retained as reasonable to bring awareness to these painful practices. Hence, the mounting scientific evidence that relates hot-iron branding to the welfare compromise of cattle [40,41,42] could guide Namibia to abolish hot-iron branding and join many countries that are increasingly prohibiting this practice [39]. Alternatively, the country could opt for microchipping for identification, or freeze branding or the use of cooling gel to reduce pain sensitivity and aid in faster brand wound healing [40,41,42]. However, the practicalities and resource availability of such alternatives may hinder their applicability. To this end, mitigating strategies for stock theft along with awareness of compromising welfare practices (e.g., dewlap cuts, ear cuts) are immediate remedial actions to be taken to improve cattle welfare at most herds in the village farms.

The threshold (10%) considered as unacceptable welfare for herds with long/sharp horns was motivated by difficulty in handling, injuries, bruises, and condemnation at slaughter [43] that are associated with horned cattle. In this study, herds with long/sharp horns had a combined derived red threshold that were more than >8 times higher than their imposed thresholds. Arguments are many for the keeping of horned cattle; for example, traditional customs and sacredness at some households [9], natural tools against predation [43,44] and for behavioural and physiological functions of cattle [43]. Thus, keeping a proportion of cows with horns, e.g., 40% (close to 38% orange threshold; Table 5) might be more appropriate as a compromise. Also, the downside of keeping the imposed 10% threshold than an adjusted 40% will be the caveats of increased disbudding of calves. Breeding for a polled state might solve the surrounding debate of keeping horned cows [45,46]. Indeed, Broom, [47] advocated for holistic critical scrutiny of issues conflicting with animal welfare standards (i.e., horned cattle), and what is accepted as normal by the farmers is not necessarily acceptable for the welfare of animals [17]. Further investigation on the impact of keeping of horned cattle against mitigating predation advantage is, thus, worth an undertaking and thresholds might be adjusted accordingly.

For *stockpersonship and handling measures*, it appears that three main issues, namely, (1) yarding frequency, (2) breed variations and (3) yard design/quality contributed to the high derived thresholds of fearful and falling/lying across the farm types. Fearfulness in cattle was associated with adverse handling [48,49] and sub-optimal production and reproductive performance [50,51]. Thus, critical analysis and understanding of the causes of high incidences of fearful cattle in the country are important. In New Zealand, infrequent yarding of beef cattle was thought accountable for the adjustment of the original imposed 2% to 5%, a more appropriate derived red threshold. This adjusted threshold was used for Namibia, but all farm types still obtained high derived red thresholds. This may indicate a need to adjust the imposed threshold, for example, to 7% (orange threshold). It is likely that the commonness of fearful/agitated behaviour may have been a temporary response to the infrequent handling and stressful environment caused by the yard’s designs and/or construction quality (see Appendix B; Figure A1). The predominant Brahman cattle were also noted to exhibit flight behaviours (as was noted by Cooke [52]), which could explain the comparable fearfulness incidences at semi-commercial and communal herds, although communal herds with more tamer crossbreds and Sanga breeds were less restrained. Brahman cattle were also noted to lie in the race more than other breeds (mostly for no apparent reason), contributing to the high derived thresholds (ratio of 6 times) of falling/lying cows across all the farm types. Like fearfulness, an adjustment in the threshold from an imposed 2% to 7% (close to 6.2% orange threshold) might be suitable as the temporary stress of handling could have likely caused cattle to lie in the yards. However, the observation of breed influence on cattle behaviour and welfare compromise demands further investigation, for possible mitigation. Cattle lying in the race contributed to others falling on top of them, which led to compromised cattle flow and more use of electrical prodders, hitting and noisy handlers.

It was important for animal-based and stockpersonship assessments to be made while cattle were handled for purposes other than the assessment itself. Thus, it might be worth describing and validating yard construction and design quality in a welfare assessment, such as to define what entails a yard, differentiate those with enough holding pens, the race and chute, as well as materials used, e.g., metal, wooden poles, tree poles and thorny bushes (see Appendix A) for the different yard designs and quality found at the farms). The definition of the yard quality and designs could bring awareness to the welfare compromise caused by poor yards [13,27] and in return encourage channelling of resources, skills and knowledge for the construction and maintenance of better standards facilities.

Finally, the study was undertaken at only 55 herds, but all herds represented the prevailing production system in the three regions. The welfare thresholds need to be validated across multiple beef herds in the country, with more assessors and consideration of the prevailing environmental conditions (e.g., seasonality and dry periods). It is hoped that this categorisation will also be validated by expert opinions to apply the correct decisions when an assessment is made, such as for on-farm benchmarking and certification of cattle or beef from such establishments. The current assessment results and thresholds could also be used as a basis for the welfare status of Namibia farms, such that, especially the village farms prioritise intervention areas of welfare compromise to either attain or maintain beef markets. This study can help to improve animal welfare in other countries with different status of animal welfare with individual thresholds of measures.

## 5. Conclusions

The study provided thresholds which could be used to categorise continuous animal welfare measures into acceptable, marginal and non-acceptable welfare, hence, providing guidance for when the intervention was needed on Namibian beef farms. Commercial farms attained welfare measures thresholds based on a New Zealand protocol—which suggest opportunities to create a system and a suitable environment that promote animal welfare in Namibia. However, the same thresholds, regardless of the system, were set for the country’s beef herds, and semi-commercial and communal herds showed less acceptable welfare of beef cows. The long-term target is to have improved derived threshold in all measures at all herds, but an initial target could be geared towards improving the worst 15% of herds. Adjusted lenient thresholds to reflect drought condition were suggested for measures related to good feeding and mortality, whilst the originally imposed thresholds were still retained under “normal conditions”. This was a good example of setting the welfare threshold for “what should be and not what is”. Nevertheless, it may be an advantage that the current thresholds were validated in the possible worst-case scenario that can hit the beef industry in Namibia. Some of the imposed thresholds (i.e., swelling, hair loss, dystocia, lameness) were retained because of the significant importance to the welfare of animals and preventiveness of the traits. Handling measures (fearful, falling/lying) and abrasions thresholds were adjusted to reflect the temporary stress caused by infrequent cattle handling, and faulty yard designs and or constructions and the possible influence of Bahaman cattle behaviour during handling. There is a need to validate this categorisation and scoring of measures on multiple farms, and with more experts to classify Namibian beef farms status correctly. It is to the advantage of Namibia to have a validated, comparable animal welfare assessment protocol based on other beef exporting countries to strive for maintaining good welfare standards across the beef production chains.

## Figures and Tables

**Figure 1 animals-11-00250-f001:**
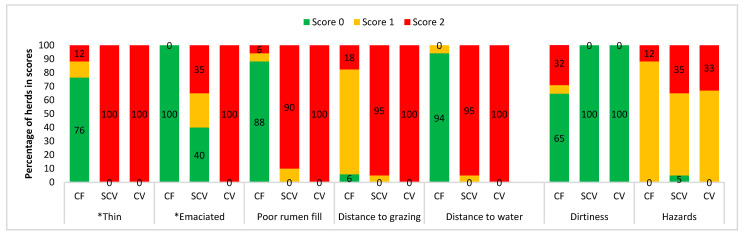
Good feeding and environment measures (separated by a gab before dirtiness) at the commercial (CF), semi-commercial (SCV), and communal (CV) beef cow herds in Namibia showing the percentage of herds per welfare scores. * Thin included all thin cows ≤2.5/5 scale, and emaciated was only cows with BCS 1–2/5 scale. See Table 1 for measures scoring descriptions and categorical ranking.

**Figure 2 animals-11-00250-f002:**
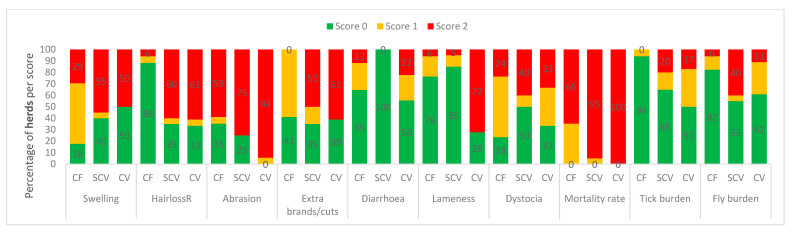
Categorised health measures at the commercial (CF), semi-commercial (SCV), and communal (CV) beef cow herds in Namibia showing the percentage of herds per welfare scores. See Table 2 for measures scoring descriptions and categorical ranking.

**Figure 3 animals-11-00250-f003:**
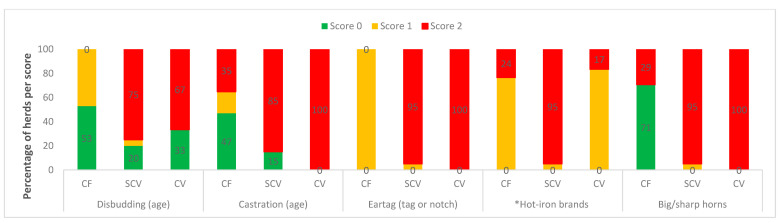
Painful management procedures at the commercial (CF), semi-commercial (SCV), and communal (CV) beef cow herds in Namibia showing the percentage of herds per welfare scores. * Hot-iron branding described one compulsory brand (score 1) or more than the one brand (score 2). See Table 2 for measures scoring descriptions and categorical ranking.

**Figure 4 animals-11-00250-f004:**
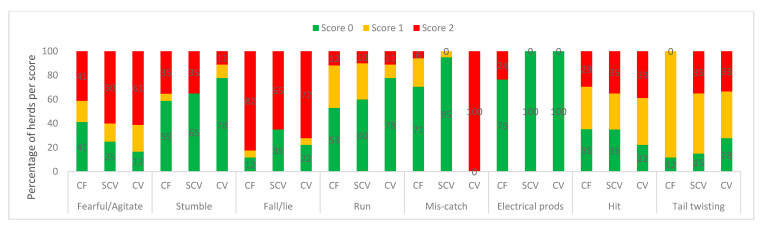
Animal based stockpersonship measures at the commercial (CF), semi-commercial (SCV), and communal (CV) beef cow herds in Namibia showing the percentage of herds per welfare scores.

**Figure 5 animals-11-00250-f005:**
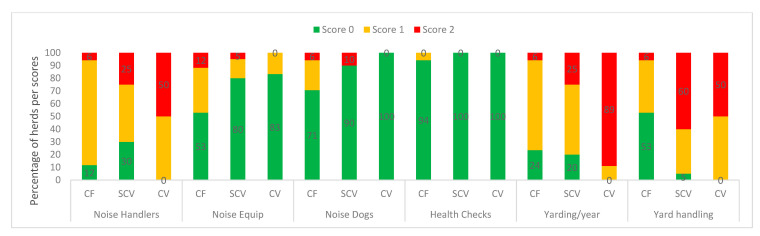
Resource-and management based stockpersonship measures at the commercial (CF), semi-commercial (SCV), and communal (CV) beef cow herds in Namibia showing the percentage of herds per welfare scores.

**Table 1 animals-11-00250-t001:** Categorical ranking of good feeding and appropriate environment measures in the proposed Namibia extensive beef cow-calf systems protocol.

Welfare Criteria	Animal Welfare Measure/Indicator	Scoring Description	Categorical Ranking
Absence of hunger	Body condition score (thin cows)	% thin/emaciated cows in herd of score (score 1–2.5) on 1–5 scale [15].	0: 0–5.0% 1: 5.1–10% 2: >10%
	Rumen fill score (poor rumen fill)	% of animals with hollow/empty rumen observed in the race [16].	0: 0–20.0% 1: 20.1–50% 2: >50%
	Distance to grazing	The questionnaire asked how far cattle had to walk to access grazing. (This included the distance to grazing for cattle that daily come to drink after grazing at water points that are provided close to yards).	0: 0–1.60 km 1: 1.61 km–3.2 km 2: >3.2 km
Absence of thirst	Distance and availability of water	Average distance to access water. (Distance to water was estimated as the distance to grazing as water points are close to yards and cattle come to drink after grazing in some herds).	0: 0–1.6.0 km 1: 1.61 km–3.2 km 2: >3.2 km
Comfort around resting	Dirty body	Total number of animals assessed as having a dirty tail, hind and flank (>25% of combined areas covered with dirt or manure).	0: 0–10.0% 1: 10.0–20% 2: >20%
Ease of movement	Absence of hazardous objects/environment	Hazardous objects observed in the yard and paddocks (i.e., sharp objects lying around, steep hills).	0: no hazards 1: 1 or 2 hazards 2: 3 or more hazards or cattle had died in any hazard

%: percentage.

**Table 2 animals-11-00250-t002:** Categorical ranking of good health measures in the proposed Namibia extensive beef cow-calf systems protocol.

Welfare Criteria	Animal Welfare Measure/Indicator	Scoring Description	Categorical Ranking
Absence of injuries/physical impairment	Abrasions, Swelling Hair loss/hairless	% of observed cows with abrasions/fresh scratches, swelling or hairless patches (>1 cm).	0: 0.0% 1: 0.1–2% 2: >2%
	Extraneous cattle markings/ branding wounds (e.g., multiple brands, dew lap cutting)	% of observed cows with brand mark wounds (>2 cm) or more than one extraneous brand mark (i.e., stock brand, initials or name of a farmer branded).	0: 0.0% 1: 0.1–5% 2: >5%
	Size and shape of horns	Number of observed cows with sharp/long horns (>5 cm in length, sharp and forward facing to pose a risk of injuring others).	0: 0–5.0% 1: 5.1–10% 2: >10%
Absence of disease	Blindness, Ocular and Nasal discharges	% of observed cows with blindness in one or both eyes. % of observed cows with ocular and nasal discharges extending 2 cm.	0: 0.0% 1: 0.1–2% 2: >2%
	Lameness	% of observed cows with gait abnormality.	0: 0.0% 1: 0.1–2% 2: >2%
	Diarrhoea	% of observed cows with evidence of diarrhoea (more than a hand wide on both sides from the base of tail).	0: 0–10% 1: >10–20% 2: >20%
	Dystocia	% of cows recorded with difficult births.	0: 0.0% 1: 0.1–2% 2: >2%
	Mortality rate	Sum of accidental deaths, deaths, due to disease, or culling because of disease/accidents in the last 12 months.	0: 0.0% 1: 0.1–2% 2: >2%
	Fly burden Tick burden	Proportion of observed cows with more than an estimated 20 flies (i.e., horse flies). Proportion of observed cows with more than an estimated 20 ticks on any part of the body of a cow.	0: 0–5.0% 1: 5.1–10% 2: >10%
Painful procedures	Disbudding Castration	Specify age at disbudding and use of anaesthetics. Specify age at castration and use of anaesthetics.	0: No disbud/castration 1: ≤2 months 2: >2 months 0: No disbud/castration 1: use of anaesthetics 2: no anaesthetics
	Ear tagging/ notching	Specify age at ear tagging and ear notching and with/without the use of anaesthetic.	0: no tag or use anaesthetics 1: tag with no anaesthetics 2: notching/cutting with no anaesthetics
	Hot-iron branding	Record branding events and the use of local anaesthetic (from the questionnaire).	0: no branding or use anaesthetics 1: one brand (compulsory) 2: more than 1 brand
	Use of electrical prodders	Estimated proportion of cows prodded with an electrical goad while drafted or standing in the race, pens or yards.	0: no prodding 1: few/occasional prod (≤1% cows) 2: many/frequent prod (>1% cows prodded)

%: percentage.

**Table 3 animals-11-00250-t003:** Categorical ranking of appropriate stockpersonship measures in the proposed Namibia extensive beef cow-calf systems protocol.

Welfare Criteria	Animal Welfare Measure/Indicator	Scoring Description	Categorical Ranking
Stockpersonship animal-based measures in and out of race	Fearful/agitated	% cows showing fearful/agitated behaviour (climbing on others or attempting to escape).	0: 0.0% 1: 0.1–5% 2: >5%
	Falling	% cows lying in or falling in race/forcing pen or on exiting.	0: 0.0% 1: 0.1–2% 2: >2%
	Stumbling	% cows stumbling while exiting to paddocks.	0: 0.0% 1: 0.1–5% 2: >5%
	Running	% cows running out of the race/holding pens into paddocks.	0: 0–5.0% 1: 5.1–10% 2: >10%
Animal handling stockpersonship and resource-based measures	Mis-catching (in chute/race)	# of cows mis-catch with gates on any part of the body either in the race or chute head bale. If no race, available mis-catch was recorded if more than one attempt was made to capture/restrain an individual animal with ropes or if a cow did not stand still when a rope was secured around the legs.	0: no mis-catch 1: mis-catch ≤1% 2: mis-catch >1%
	Hitting	% of cows hit or poked with moving aids.	0: no hitting 1: occasional/few hit 2: frequent hit/poke (>10% cows)
	Tail twisting	Estimate the proportion of cows with tail twisted while drafted or standing in the race or pens.	0: no twisting 1: occasional/few twist (≤10% of cows) 2: frequent twist (>10% of cows)
	Noise of handlers Noise of equipment/ machinery Dogs noise around the yard	Evaluate noise of handlers, noise of equipment (race or chute gate) and machinery (generators etc.) and observe the presence and noise frequency of dogs around the yard.	0: no noise/dogs 1: minor audible/occasional noise 2: unpleasantly/persistent noisy handlers/equip/dogs
	Health checks	Frequency of health checks on cows during pregnancy.	0: daily 1: once-twice/week 2: more than weekly
	Yarding frequency	Frequency of yarding of cows per year for handling or restraining.	0: >4 times 1: 3–4 times 2: 0–2 times
	Yard flow/handling of cattle	Yard flow of cattle influenced by handling facilities design/quality.	0: very effective cattle flow 1: effective but with flaws 2: difficult flow

%: percentage. #: proportion.

**Table 4 animals-11-00250-t004:** The ratio of red thresholds (derived from data) over the imposed categorisation thresholds of the individual 3-beef cow-calf farming systems in Namibia. Derived thresholds were determined based on z scores to result in approximately 50% of herds falling into an acceptable welfare band (‘green’) and 15% of herds into a poor welfare band (‘red’). Herds not in the green or red band were classified as orange”.

Measures	Imposed Categorisation Thresholds (%)	Commercial (*n* = 17)				Semi-Commercial (*n* = 20)				Communal (*n* = 18)			
		Mean(%)	Orange Threshold	Red Threshold	Ratio of Threshold	Mean (%)	Orange Threshold	Red Threshold	Ratio of Threshold	Mean (%)	Orange Threshold	Red Threshold	Ratio of Threshold
# Thin cows	10	7.2	2.5	11.6	1.2	70.7 ⁿ	65.6	102.6	10.3 *	97.7 ⁿ	97.6	102.7	10.3 *
# Emaciated cows	10	0.0	0.0	0.0	0.0	11.1 ⁿ	5.4	23.8	2.4 *	82.9 ⁿ	81.8	98.0	9.8 *
Poor rumen fill	50	4.4	1.2	6.3	0.1	49.1 ⁿ	44.7	72.9	1.5	76.2 ⁿ	74.3	95.7	1.9
Dirtiness	20	0.7	0.3	1.5	0.7	0.0	0.0	0.0	0.0	2.0	0.8	3.6	0.2
Swelling	2	2.1	1.6	3.8	1.9	4.9	2.2	9.1	4.6 *	3.3	1.7	6.7	3.3 *
Hair loss	2	0.3	0.1	0.7	0.3	3.6	1.9	7.2	3.6 *	6.8	3.5	14.9	7.5 *
Abrasion	2	3.2	1.9	6.6	3.3 *	8.7	4.8	18.9	9.5 *	20.4 ⁿ	16.9	35.7	17.8 *
Extraneous brands/cuts	5	1.0	0.8	2.1	0.4	8.1	3.0	15.1	3.0 *	10.7	4.0	19.1	3.8 *
Long/sharp horns	10	7.2	3.1	13.6	1.4	45.5 ⁿ	40.1	70.6	7.1 *	59.3 ⁿ	54.9	88.3	8.8 *
Blindness	2	0.3	0.2	0.8	0.4	0.0	0.0	0.0	0.0	0.1	0.0	0.3	0.1
Ocular discharge	2	0.1	0.1	0.3	0.2	0.0	0.0	0.0	0.0	0.1	0.0	0.3	0.1
Nasal discharge	2	0.1	0.1	0.3	0.2	0.0	0.0	0.0	0.0	0.1	0.0	0.3	0.1
Diarrhoea	10	3.4	1.7	6.9	0.7	0.1	0.1	0.3	0.0	7.1	3.5	14.9	1.5
Lameness	2	0.5	0.3	1.1	0.6	0.3	0.1	0.7	0.4	5.1	2.8	10.5	5.3 *
Dystocia	2	0.9	0.7	1.7	0.8	2.2	1.2	4.5	2.3 *	1.9	1.3	3.8	1.9
Tick burden	10	0.4	0.1	0.9	0.1	11.8	2.6	17.3	1.7	6.0	2.8	12.2	1.2
Fly burden	10	1.8	0.6	3.2	0.3	14.1	3.5	25.2	2.5 *	9.5	3.3	17.2	1.7
Mortality rate	2	3.4	2.8	5.4	2.7 *	12.3 ⁿ	9.5	23.9	11.9 *	13.2 ⁿ	10.7	22.7	11.4 *
Fearful/Agitate	5	6.2	3.4	12.7	2.5 *	7.1	4.5	14.8	3.0 *	7.4 ⁿ	5.2	15.6	3.1 *
Fall/lie	2	6.5 ⁿ	4.8	12.9	6.4 *	6.4	2.7	12.7	6.4 *	5.7 ⁿ	3.8	12.0	6.0 *
Stumble	5	3.8	1.8	7.5	1.5	3.9	1.2	6.2	1.2	1.8	0.6	3.2	0.6
Run exit	10	5.6	3.5	11.2	1.1	3.3	1.2	6.2	0.6	3.1	1.4	6.0	0.6

ⁿ Measures were normally distributed. * Measures had the ratio of derived red threshold: Imposed threshold that was >2 times. # Thin cows included all cows with BCS ≤ 2.5/5 scale, and emaciated was only the proportion of thin cows with the severity of BCS 1–2/5 scale.

**Table 5 animals-11-00250-t005:** Combined ratio of red thresholds (derived from data) over the imposed categorisation value of the 3-beef cow-calf farming systems in Namibia (*n* = 55). Derived thresholds were determined based on z scores to result in approximately 50% of herds falling into an acceptable welfare band (‘green’) and 15% of herds into a poor welfare band (‘red’). Herds not in the green or red band were classified as orange”.

Measures	Imposed Categorisation Threshold (%)	Mean (%)	Orange Threshold (%)	Red Threshold (%)	Ratio of Red Thresholds/Imposed
# Thin cows	10	59.9	29.4	166	16.6 *
# Emaciated cows	10	31.2	0.9	62.4	6.2 *
Poor rumen fill	50	44.2	20	121.8	2.4 *
Dirtiness	20	0.9	0.3	1.6	0.9
Swelling	2	3.5	1.7	5.2	2.6 *
Hair loss	2	3.6	1.3	6.6	3.3 *
Abrasion	2	10.6	6.5	25.2	12.6 *
Extraneous brands/cuts	5	6.8	2	8.9	1.8
Long/sharp horns	10	38.2	14.7	82	8.2 *
Blindness	2	0.1	0.1	0.4	0.2
Ocular discharge	2	0	0	0.2	0.1
Nasal discharge	2	0	0	0.2	0.1
Diarrhoea	10	3.4	1.3	6	0.6
Lameness	2	1.9	0.7	3.4	1.7
Dystocia	2	1.7	0.7	3.3	1.7
Tick burden	10	6.4	1.6	8.9	0.9
Fly burden	10	8.8	2.2	13.5	1.4
Mortality rate	2	9.8	6.9	18.1	9.1 *
Fearful/Agitate	5	6.9	4.3	14.2	2.8 *
Fall/lie	2	6.2	3.6	12.7	6.4 *
Stumble	5	3.2	1.2	5.4	1.1
Run exit	10	3.9	1.8	7.7	0.8

* Measures had the ratio of derived red threshold: Imposed threshold that was >2, # Thin cows included all cows with BCS ≤ 2.5/5 scale, and emaciated was only the proportion of thin cows with the severity of BCS 1–2/5 scale.

## Data Availability

All relevant data were analysed and reported in Part 1 (https://pubmed.ncbi.nlm.nih.gov/33445688/) or the current Part 2 of the study.

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
