# Peer review of "Assessing Extensive Semi-Arid Rangeland Beef Cow-Calf Welfare in Namibia. Part 2: Categorisation and Scoring of Welfare Assessment Measures"

_animals, 2021, doi:10.3390/ani11020250_

Round 1

Reviewer 1 Report

A good first project for an important area.

129-131: there should be more justification for the author's opinion.  explain why the author came to this conclusions.

159-199: as most of this was presented in the "Part 1" paper at the most a table highlighting the results previously put should be given.

Table 4: use the terms Score rather than red / orange. Explain what the term means in a footnote to the table.

Tables 4 and 5: explain why the levels of BCS used varies in these table.

I'm not sure there is justification for adjusting the thresholds during drought conditions.  

The issues relating to ectoparasites is species (of the parasite and the host) specific.  Is there any literature describing the endemic species?

A short comment on the practicalities of alternative ID practices e.g. freeze branding, microchipping could be added.

Appendix A: the photos should be individually labelled.

Author Response

Great review! Thanks.

Reviewer 2 Report

My main concern is about objectivity of scoring. I guess you have to add some sentences to explain weakness of scoring in such an imminent field.

Line

Comment

115

Significance kevel?

200

Each axis has to be explained and described by name in the figure, as figure 2

260

This part suggests an objectivity which doesn’t exists due to the subjective scoring by the person giving the scores. If you would have 3 persons scoring and would have taken the mean for each score it would be rather objective. But you have not done so. No problem. The problem is imminent. But you have to discuss this issue at this point.

290

Compared to BCS, RFS is a short-term score. This has to be added to your discussion.

327

Why don’t you compare New Zealand scoring with Namibia scoring in an additional table, where it can be fitted? It is not against Namibia, but it would show where are the highest differences and where are the targets for improvement.

426

You can also write this study can help to improve animal welfare in other countries with different status of animal welfare with an individual threshold.

509

8065-8073

523

check font

544

check font of link and point at the end

561

check points

563

check pages

569

check pages

571

number?

Author Response

Great review! Thanks so much.

Round 2

Reviewer 2 Report

Thank You for addressing all my comments minus one.